# Health system opportunities and challenges for PrEP implementation in Kenya: A qualitative framework analysis

Kaitlyn Atkins[1]*, Abednego Musau[2], Mary Mugambi[3], Geoffrey Odhyambo[2], Soud Ali Tengah[2], Mercy Kamau[2], Ruth Kumau[2], Jason Reed[4], Daniel Were[2]

1 Social and Behavioral Interventions Program, Department of International Health, Johns Hopkins Bloomberg School of Public Health, Baltimore, Maryland, United States of America, 2 Jhpiego Kenya, Nairobi, Kenya, 3 National AIDS Control Program, Kenya Ministry of Health, Nairobi, Kenya, 4 Jhpiego Corporation, Baltimore, Maryland, United States of America

* kait.atkins@jhu.edu

**Data Availability Statement:** Data are available in Supporting Information and at this link: https://clinepidb.org/ce/app/workspace/analyses/DS_d70aacce42/new/details.

## Abstract

### Background

As pre-exposure prophylaxis (PrEP) scales up in sub-Saharan Africa, governments and implementers need to understand how to best manage national programs. Kenya's national PrEP program offers an opportunity to review elements of program success within the health system and evaluate the utility of a national implementation framework. We explored health system considerations for PrEP implementation to understand how Kenya's national PrEP implementation priorities align with those of PrEP service providers, peer educators, and program or county managers.

### Methods

We conducted twelve key informant interviews (KII) and nine focus group discussions (FGDs) with PrEP program and county managers (n = 12), peer educators (n = 44), and PrEP service providers (n = 48). We recruited participants across a variety of cadres and experiences with PrEP programs. KIIs and FGDs focused on PrEP service delivery and program implementation. Data were collected by trained study staff, audio recorded, translated into English, and transcribed. We used framework analysis methods to systematically apply Kenya's 2017 National PrEP Implementation Framework to the data and summarized findings according to the seven Implementation Framework domains.

### Results

All respondents emphasized the important role of communication, coordination, training, and leadership in PrEP implementation. PrEP service providers and program and county managers highlighted the importance of efficient data collection and utilization, and improved resource allocation. Commodity security and research, while key elements of the PrEP Implementation Framework, were less commonly discussed, and research was less

**Funding:** This work was supported, in whole or in part, by the Bill & Melinda Gates Foundation [Investment ID: INV-007340]. Under the grant conditions of the Foundation, a Creative Commons Attribution 4.0 Generic License has already been assigned to the Author Accepted Manuscript version that might arise from this submission. In addition, the project received a donation of TDF/FTC from Gilead Sciences. The funders had no role in study design, data collection and analysis, decision to publish, or preparation of the manuscript.

**Competing interests:** The authors have declared that no competing interests exist.

prioritized by respondents. Respondents highlighted the importance of coordinated PrEP service delivery across sites and programs to improve overall client experiences.

## Conclusion

In the context of a nationally-scaled PrEP program, PrEP service providers, peer educators, and program and county managers value strong leadership, close coordination of services across sites, and expedient use of data to improve strategies and services. Kenya's PrEP Implementation Framework aligns closely with the priorities of individuals involved in PrEP service delivery and management, and provides a comprehensive overview of health system considerations for effective implementation of a PrEP program at scale.

## Introduction

In recent years, Kenya has made substantial progress in curbing the HIV epidemic through scale-up of testing, treatment, and prevention programs [1]. However, in 2018 nearly 36,000 adults were newly-infected with HIV [2]. The greatest number of new infections occurred among men and women aged 15–24 years and key populations, including female sex workers (FSW), men who have sex with men (MSM), and people who inject drugs [3].

Pre-exposure prophylaxis (PrEP) to reduce HIV acquisition risk is a critical component of a comprehensive HIV response. In 2015, in response to evidence from four randomized controlled trials [4–7], the World Health Organization recommended the use of tenofovir disoproxil fumarate and emtricitabine for PrEP for individuals at substantial risk of HIV infection [8]. Shortly thereafter, the Kenyan government approved oral PrEP for HIV prevention, and in 2016 Kenya included oral PrEP in its national antiretroviral therapy guidelines [9].

National PrEP scale-up in Kenya was launched in May 2017 [10]. In sub-Saharan Africa (SSA) more broadly, as of December 2020, PrEP programs are ongoing in 23 countries, with 18 countries implementing PrEP at scale (i.e. outside the context of research or demonstration project) [11]. A recent review of PrEP roll-out in Africa highlighted that, while there is tremendous interest in PrEP, programs would benefit from simplification of services, enhanced provider capacity-building, and improved PrEP access and convenience [12]. Indeed, as PrEP programs continue to expand throughout SSA, it is imperative that governments consider how to best structure and organize national programs in order to ensure eligible clients are supported to access and continue using PrEP. Without strengthened health systems, significant access and uptake of PrEP is unlikely to be realized. Well-functioning health systems improve population health, provide social protection, respond to legitimate expectations of citizens, and contribute to economic growth [13, 14]. Further, it is an ethical imperative for health care providers to offer PrEP to those who might benefit, and for health systems to implement standards supporting routine provision of PrEP [15]. However, as of December 2020 only ten country programs were guided by implementation plans or frameworks, and Kenya was one of only three countries in SSA with a dedicated PrEP service delivery or implementation policy framework [11].

Multiple recent reviews have outlined considerations for PrEP implementation and scale-up in SSA [12, 16], but primary research literature around integrating PrEP into health systems in the region remains sparse. Further, there are gaps in understanding frontline health care workers' experiences with PrEP implementation. Several studies have prospectively examined health care workers' views on PrEP as a hypothetical HIV prevention intervention in SSA and elsewhere [17–19], but few have directly engaged health care workers who are experienced in

PrEP implementation. Existing research with PrEP service providers has highlighted the importance of health care worker perspectives in considering implementation challenges and opportunities [20, 21]. In the COVID-19 pandemic context, which has seen overburdened health systems [22], closures of medical facilities, and increased burnout in health care worldwide [23, 24], it is even more critical to understand the expectations, experiences, and suggestions of frontline workers who routinely provide PrEP services and to consider how the pandemic and post-pandemic contexts may impact PrEP services at a health system level [25].

Kenya's nationally-scaled PrEP program offers an opportunity to review successes and challenges of scaling PrEP within the health system. Here, we examine the perspectives of program and county managers, peer educators, and PrEP service providers, using Kenya's national PrEP Implementation Framework to identify opportunities for PrEP program strengthening that can be applied in health systems throughout the region.

## Methods

### Study setting

This analysis was part of a larger evaluation of *Jilinde*, a five-year project led by Jhpiego and partners in Kenya since 2016. Currently, *Jilinde* supports PrEP services in ten of the Kenyan government's 19 priority counties in Kenya [26]. *Jilinde's* goal is to demonstrate the feasibility of PrEP implementation at population scale and in low resource settings. Working in partnership with the Government of Kenya, *Jilinde* supports PrEP service delivery through private and public health facilities and drop-in centers (DICEs) that primarily provide HIV services to key populations. The program aims to reach individuals who are most vulnerable to HIV infection, with a specific focus on MSM, FSW, and adolescent girls and young women (AGYW).

### Implementation framework development

Public sector implementation of PrEP in Kenya is guided by the *Framework for the Implementation of Pre-Exposure Prophylaxis of HIV in Kenya*, or PrEP Implementation Framework (PIF) [26]. The PIF was developed in 2017 by Kenya's National AIDS and STI Control Program (NASCOP). Development of the PIF was a consultative process between the Ministry of Health through NASCOP and implementing partners, donors and relevant community stakeholders through the stewardship of a national PrEP Technical Working Group. The PIF guides policymakers, government officials, donors, program managers, PrEP service providers, and others on delivery of PrEP in Kenya. It outlines seven key focus areas for PrEP implementation: (1) Leadership and Governance; (2) Service Delivery; (3) Commodity Security and Supply Chain Management; (4) Communications, Advocacy, and Community Engagement; (5) Monitoring and Evaluation; (6) Research and Impact Evaluation; and (7) Financing and Resource Mobilization.

### Data collection

For this qualitative analysis, we collected data cross-sectionally from peer educators, PrEP service providers, and program and county managers (Table 2). Participants were recruited through a purposive sampling strategy. We recruited providers from their respective health facilities via invitations sent to facility in-charges. Invitations mentioned the number of providers from various roles to be invited from the facility and included a recruitment script. PrEP service providers included nurses, clinical officers, HIV testing services providers, and other cadres actively engaged in PrEP delivery or working in the facility's HIV program. Peer

educators were affiliated with DICEs or public facilities providing PrEP and were recruited by the respective facility in-charges. Key informants were individuals holding management positions in DICEs or at sub-county health management teams in *Jilinde*-supported counties. We conducted nine focus group discussions (FGDs), five with PrEP service providers and four with peer educators, and twelve semi-structured key informant interviews (KIIs) with program and county managers.

For all KIIs and FGDs, we used standard interview and discussion guides with a set of core questions and probes (S1 File). We iteratively modified guides over the course of data collection to address emergent issues related to PrEP implementation.

We conducted KIIs (n = 12) between November 2018 and March 2019. Interviews broadly assessed community perceptions of HIV, acceptability of PrEP, service delivery and access issues related to PrEP, prioritization of PrEP as an intervention, and communications considerations for PrEP service delivery. We also asked stakeholders about specific successes and challenges faced by their organization in providing PrEP, and for feedback about their partnership with the *Jilinde* project.

We conducted PrEP service provider FGDs (n = 5) between April 2018 and February 2019. FGDs assessed provider acceptability of PrEP, community norms around PrEP, and experiences providing PrEP. We conducted peer educator FGDs (n = 4) between November 2017 and July 2019. These FGDs assessed peer educators' perceptions of HIV prevention technologies broadly, community acceptability of PrEP, demand generation, information gaps around PrEP, and considerations around PrEP access. We also asked peer educators about their motivating factors and any challenges they faced in their day-to-day work mobilizing their peers for PrEP and other HIV services.

All KIIs and FGDs were conducted by trained qualitative researchers, and were held in private locations. Provider FGDs were conducted in English; peer educator FGDs and KIIs were conducted in Kiswahili with direct field translation into Dholuo as needed. We compensated participants Ksh 500 (~USD 5) for transportation costs. All data were audio recorded, transcribed, and translated into English. FGDs and KIIs were conducted in private locations.

## Data analysis

Our analysis was guided by the Framework Method [27], a systematic method for qualitative analysis which uses matrices to structure data reduction and analysis. This structure is then used to compare data across and within cases. We used the PIF [26] to structure our analysis.

After transcribing and translating the data, we first read each transcript to gain a holistic sense of the KII or FGD as a whole. We then used open coding on a selection of transcripts from each group to ensure the PIF captured all important aspects of data before developing a codebook based on the PIF (Table 1). We used sub-codes to examine key elements of each PIF focus area. We applied the codes and sub-codes to the data in Dedoose, and then charted the data into a framework matrix, which contained summaries of the data from each transcript by code/subcode [27]. Finally, after charting the data, we reviewed the matrix to identify patterns within and across data. Ultimately, due to overlap across sub-codes, we summarized patterns across each key PIF focus area (parent code) rather than at the granular sub-code level. Throughout the data analysis process, we kept notes of impressions, concerns, ideas, and emerging themes; these were continually revisited in team meetings to inform analysis.

## Ethical considerations

We received ethics approval from the Institutional Review Boards of the Kenya Medical Research Institute (KEMRI; Non-KEMRI 601) and from the John Hopkins School of Public

**Table 1. Summary of codes.**

| Parent Code (PIF Focus Area) | Sub-Codes |
|---|---|
| Leadership and Governance | Facility management; provider leadership; community leadership |
| Service Delivery | Identifying and initiating PrEP users; providing PrEP; clinical monitoring and follow-up; training staff; ensuring and improving quality |
| Commodity Security and Supply Chain Management* | Supply, dispensing and distribution; reporting |
| Communications, Advocacy, and Community Engagement | Demand generation; communication needs; advocacy; community engagement |
| Monitoring and Evaluation | Facility M&E processes; data collection tools; M&E needs |
| Research and Impact Evaluation† | Opportunities for further research; research management and coordination |
| Financing and Resource Mobilization | User financing; facility financing; community financing |

* This code was expanded beyond the original PIF focus area (commodity security) to include broader issues of supply chain management, which were more commonly discussed during data collection.
† This PIF area summarizes a research and evaluation agenda for PrEP implementation. While it was included as a code during analysis, its purpose was to highlight areas where additional research may be warranted. As such, it is not included in the Results below. Areas for future research are included in the Discussion.

Health (JHSPH; IRB No. 7657) to conduct this research. All participants provided verbal and written informed consent before participating in the study. In the written consent, participants were given the option to omit their names but only provide a signature in an effort to safeguard their identity. For FGDs, participants were asked not to use their own names or other identifiable information during the discussion.

# Results

Our final sample included 104 participants. Participants' median age was 29 years (interquartile range = 12), 66% were female, and nearly half were from Lake Region (Table 2). In total, our sample included 48 PrEP service providers, 44 peer educators and 12 program and county managers. Participants represented seven of the ten *Jilinde*-supported counties: Kiambu, Kisumu, Kisii, Machakos, Migori, Mombasa, and Nairobi.

We identified themes within each of the seven PIF domains. Overall, we found that respondents across groups emphasized the important role of communications, training, and leadership in PrEP implementation. PrEP service providers and program and county managers highlighted the importance of efficient data collection and utilization and improved resource allocation to fill gaps. Respondents felt that harmonization of processes (e.g. risk screening, drug dispensing, client flow) across PrEP sites and programs would improve PrEP service delivery and enhance client experiences by making it easier to navigate services.

## Leadership and governance

All respondents highlighted the importance of strong leadership, both at a national level and within individual programs. Across groups, respondents emphasized that the day-to-day successes and challenges of PrEP implementation were highly dependent on management of programs and facilities. At the program level, management strategies varied across settings, and some respondents discussed challenges while others said they were highly satisfied with program leadership. Several respondents felt encouraged by strong government commitment to

the PrEP program, as signaled by the establishment of the national PrEP Technical Working Group and efforts across government levels to support PrEP implementers.

Between sites and governing bodies, many respondents discussed the need for streamlined coordination. A 33-year-old female PrEP provider from a DICE noted that at times they received conflicting messaging from government authorities and program leadership. A 40-year-old female county manager suggested that collaboration between government agencies and implementing partners could include collective data review meetings and joint supportive supervision, along with "joint planning such that . . . the government proposes this, the partner proposes that, and they agree a way forward." Program and county managers emphasized the need for strong "partnership relationships" between implementers and government agencies. Others emphasized the value of collaboration between implementing organizations:

> "One of the things I liked most is when we meet with other CSOs. The sharing of the good practices and bad practices. Actually, the best thing was that you're not alone. We are all struggling. Learning from others is very useful."– 50-year-old female Program Manager, Nairobi-based CSO

**Table 2. Summary of participant demographic characteristics (n = 104).**

| Characteristic | | No. (%) |
|---|---|---|
| **Age in categories** | | |
| | 24 years and below | 26 (25.2%) |
| | 25–34 years | 51 (49.5%) |
| | 35 years and above | 26 (25.2%) |
| | Missing | 1 (0.1%) |
| **Biological sex** | | |
| | Male | 35 (33.7%) |
| | Female | 69 (66.3%) |
| **Occupation** | | |
| | Peer mobilizer | 44 (42.3%) |
| | PrEP service provider | 48 (47.1%) |
| **Geographic region** | | |
| | Lake | 48 (46.1%) |
| | Nairobi | 27 (26.0%) |
| | Coast | 29 (27.9%) |
| **Participant type** | | |
| | PrEP service provider* | 48 (46.1%) |
| | Peer educator† | 44 (42.3%) |
| | Program and county manager‡ | 12 (11.5%) |

* Providers were recruited from Machakos, Migori, Mombasa, and Nairobi counties into 5 FGDs. They included clinicians (nurses, clinical officers, and HIV testing services providers) involved in PrEP service delivery. We included a mix of providers from *Jilinde*-supported sites and non-supported sites, and representation of providers across cadres.

† Peer educators were recruited from Mombasa and Migori counties into 4 FGDs. They included trained and sensitized community members who partner with *Jilinde* supported sites to raise awareness of PrEP at community hotspots, link individuals to PrEP sites for services, and follow up with PrEP clients.

‡ Managers were recruited from Kisii, Migori, Mombasa, Nairobi, Kiambu, and Kisumu counties for 12 KIIs. They included county and sub-county health managers responsible for HIV services and managers of civil society organizations (CSOs) contracted by Jilinde to either generate demand for PrEP or provide key population- or adolescent-friendly PrEP services in DICEs.

Beyond collaboration between the government and individual programs, some respondents, particularly PrEP service providers and peer educators, discussed a need for increased prioritization of PrEP programs within sites. Providers discussed how a lack of institutional support for PrEP programs may impact service quality, waiting times, and staff burnout. Some felt these issues could be resolved with facility-level changes, while others felt that limited support for PrEP programs were more emblematic of an overall prioritization of treatment over prevention in the health system. For example, providers discussed how provision of antiretroviral drugs for the treatment of HIV was often prioritized because leadership felt treatment outcomes had more severe implications than PrEP:

"If you miss antiretrovirals then you will have a high viral load and you are still going to be a burden. But if you miss PrEP probably [people perceive] there is nothing that will happen."– 33-year-old female clinical officer, Machakos

Within sites, respondents across groups, including PrEP service providers and peer educators who did not hold formal leadership positions, expressed a strong commitment to their own role in PrEP leadership. Many providers expressed a desire to be involved in more on-the-ground decisions, and peer educators valued the opportunity to serve as "PrEP champions" in their communities. Moreover, many respondents said they saw their own leadership and commitment to PrEP as critical for program success:

"If, as a health care provider, I know this thing [PrEP] is effective and I can tell confidently it is effective to this percentage, when I am issuing PrEP I will be very confident and I will provide a lot of support to this client. Unless the health care workers themselves believe in the drug, then the client will not believe it."– 31-year-old male nurse, Machakos

## Service delivery

**Providing PrEP.**   Service delivery platforms varied across respondents' programs, and included delivery of PrEP through DICEs, outreaches, sexual and reproductive health clinics, and comprehensive care clinics (CCC) for HIV treatment (Table 3).

While they discussed benefits and challenges to each approach, most respondents felt that offering dedicated PrEP services through outreaches, community safe spaces, and DICEs was preferable to delivery through CCC, where PrEP clients would be served alongside people living with HIV. Respondents expressed that delivering PrEP through CCC, while streamlined

**Table 3.  PrEP service delivery platforms discussed by respondents.**

| Service Delivery Platform | Description |
|---|---|
| Community outreaches & safe spaces | PrEP is provided at accessible locations during convenient hours. Locations are typically spaces in the community which have been identified as safe and friendly for key and vulnerable populations to access health services. |
| Comprehensive care clinics for HIV treatment (CCC) | PrEP is provided for HIV negative individuals through clinics that offer comprehensive treatment and follow-up services for HIV positive individuals. |
| Drop-in centres (DICE) | PrEP is offered through standalone centres in a friendly and nonjudgmental way as part of combination HIV prevention services targeting key populations. |
| Sexual and reproductive health (SRH) clinics | PrEP is provided alongside routine SRH services, including at antenatal and postnatal care, STI and family planning clinics. Target populations typically include women. |

from a staffing and commodities perspective, was not client-friendly, except when serving serodiscordant couples. A 44-year-old male county manager explained, "Sometimes they may be stigmatized. . . they don't feel free with the infected people who are coming for the antiretrovirals." Client wait times were also a concern for many respondents, particularly when it came to integrating PrEP with other HIV or sexual and reproductive health services; the same county manager said that clients "would be pretty happy to spend less time in the facilities."

Where PrEP was delivered alongside other services, program and county managers differed in their views on how to allocate PrEP roles amongst staff in facilities. A 55-year-old male director of a Migori-based CSO stated about peer educators, "I don't want them just to be labelled as the PrEP demand creators." In contrast, many PrEP service providers felt that it was important to have dedicated PrEP staff, to ensure programs are adequately supported and staff are not overextended:. A 32-year-old female nurse from Machakos explained, "so that we don't miss patients on PrEP, we identified a PrEP focal person . . . so if the HIV testing counsellor gets a patient who needs PrEP, she knows where to take the patient and who to contact." In programs where PrEP was offered alongside other services, respondents emphasized the importance of close coordination and communication between PrEP programs and other HIV services in the same facility.

Many respondents expressed that it was challenging to streamline service delivery in the face of broader health system challenges, such as staff turnover and shortages. Staff shortages had implications for PrEP service providers, but ultimately respondents said they influenced their ability to provide services to clients. For example, a 22-year-old female HIV testing services provider from Migori explained that in their facility, "we have insufficient nurses there. The one who provides family planning is still the one providing PrEP. Sometimes the youths who come for PrEP go without PrEP, since there is no person to provide it for them." PrEP service providers frequently discussed feeling burned out and overextended, due to limited staff in PrEP programs and in facilities more broadly:

> "Another thing which actually contributes to those issues for instance in our case is work load. We have one clinician who is supposed to do the outpatient, is supposed to clear the queue, at the same time is the person who is supposed to operate in the CCC. At times he gets tired and tells them you come tomorrow for PrEP [participants laughing]."– 29-year-old male social worker, Machakos

**Training staff.**   To improve service delivery, beyond broader health system changes, several respondents discussed the importance of targeted staff capacity-building. Respondents across groups expressed that, while program staff were very aware of PrEP, there were training gaps for non-clinical program staff, such as receptionists or cleaners. One program manager described how, after holding a single training, "I realized PrEP needed some constant training." The manager further described the importance of repeated peer educator training given their central role in the program:

> "The [peer educator] is the client's first contact. If you give wrong information to clients there [in the community], if someone comes here [to the facility], I don't know whether providers will be ready."– 34-year-old Program Manager, Nairobi-based CSO

Others discussed the importance of training PrEP service providers and peer educators to help potential clients make informed decisions about PrEP without feeling like they were forced to take it because of a certain identity or behavior:

"We need to create a very supportive environment to ensure that something like PrEP is really available and can be provided in a very friendly way, not in a way where you are pushing it down someone's throat. . . . You should take PrEP because according to you, you feel that is what I need [not because you are deemed to be "high-risk" by a provider]." -- 39-year-old male program manager, Kisumu-based CSO

Indeed, across groups, respondents said that high-quality PrEP training would not only ensure accurate information is given to clients, but would also ensure friendly services were provided to clients.

## Commodity security and supply chain management

Generally, only a few respondents expressed issues related to commodity supply and security. Some respondents referenced a national stock-out of PrEP commodities in 2018, and a few individual programs experienced PrEP stock-outs, but they said these were short-lived. However, several respondents noted challenges with supply of non-pharmaceutical commodities:

We get commodities from KEMSA (Kenya Medical Supplies Authority) but still with a few non-pharms we get challenges. Some partners have been supporting us with a few non-pharms like gloves, cotton wool. . . The county is able to support drugs fully–with the non-pharms we get partners supporting, but the other commodities we get from KEMSA.– 44-year-old male county manager

This manager highlighted that, while much attention is given to PrEP medicine in the supply chain, other critical commodities are often left out. Other program and county managers shared similar challenges, explaining that procurement of PrEP commodities through KEMSA was relatively straightforward, but that their program experienced stock-outs of other commodities, like HIV test kits, which are essential for PrEP services. A 34-year-old male manager of a Kiambu-based CSO noted that they felt concerned about offering PrEP when they were unable to provide condoms and lubricants which are also essential to PrEP clients.

Peer educators, though not involved in direct distribution of PrEP commodities, were also aware of and concerned about stock-outs, saying that when clients experienced a stock-out, they may be less likely to come back for PrEP. A 21-year-old female peer educator from Migori described how stock-outs could result in missed clients, saying, "What they [clients] don't like is that when they go to the facility and find the drug is out of stock and are told to come back another day. So they change [their minds]."

For commodity distribution, respondents across groups said that DICEs were ideal sites for key populations, compared to public health facilities. For example, according to a 22-year-old female peer educator from Mombasa and her fellow FGD members, clients who picked up their drugs at public health facilities "always got trouble" and encountered staff who "were not well sensitized," while in DICEs staff were perceived to "know what to do and who should get the drugs." PrEP service providers also felt that DICEs were more client-friendly.

Some respondents suggested that a "one-stop shop" system wherein clients are screened, enrolled, and pick up PrEP from the same provider would be preferable, if coordination with pharmacists was prioritized:

"One day we had a discussion with the Jilinde people and the pharmacist was there . . . We came to an agreement that, I can prescribe then I go with that prescription to the pharmacist so that she can account for that drug. But I can pick the drug, then I go with it, then I

dispense to my clients. It's just a matter of discussing it together."– 28-year-old female clinical officer, Migori

[Moderator: What else can we do?]

"In my facility I'm dispensing medication in the same way. Clients comes, he's attended to, dispensing medication there . . . Then at the end of the day, you carry the medicine that is remaining, plus what [information] is supposed to be captured, then it is entered in the pharmacy records. It can be as simple as that to make it easy for the clients."– 41-year-old female clinical officer, Migori

One program manager suggested that such a system could even involve bringing PrEP medication, condoms, lubricant, and other commodities to outreach. Despite their collective enthusiasm for "one-stop" commodity distribution systems, respondents expressed concerns that facility layouts, time constraints, and coordination across units may present challenges in implementing such a system.

## Communications, advocacy, and community engagement

**Demand generation.** Respondents shared successful strategies for demand creation, and expressed the importance of using creative approaches for reaching clients with PrEP. Respondents reported that programs relied heavily on peer educators for demand creation, and peer educators themselves felt confident and well-trained to generate demand and communicate with PrEP clients.

Some respondents expressed a need for sites and PrEP service providers to improve coordination with peer educators. In addition, respondents suggested providers could improve their communication with PrEP clients. Providers themselves expressed a desire to feel more confident in the information they were sharing with clients. For example, one 29-year-old male clinical officer from Machakos emphasized that "adhering to a clinic appointment doesn't really mean you adhere to the drugs," and that "the first information you give to that client really matters for whether he or she will continue taking PrEP." Another provider from Migori county described the need to be well-versed in PrEP, saying, "We should not sit and wait for the clients to come and ask us, but we should be ready for any client that comes." A 33-year-old male program manager from a private facility in Kisumu noted that clear provider communication could help clients "make a full decision" to use PrEP.

In generating demand, respondents felt there was a need for more tailored messaging around PrEP. One 44-year-old male county manager emphasized the need for tailored messaging for key populations, saying that "there is a lot of awareness, but still people think they are not at a substantial ongoing risk to benefit from PrEP." At the same time, some respondents were hesitant that key population-focused messaging could cause negative impacts:

"Let's not generalize, it's not good for everybody . . . don't say 'it was proposed for sex workers,' let them give you their view. They might tell you 'we don't want that.' Let's not assume . . . let's bring the people on the table. Let's not think because they have not gone to school they don't understand or they don't know why it is good." -- 50-year-old program manager of a predominantly FSW-serving CSO

**Community engagement.** Indeed, respondents felt that overly-targeted communications could result in perceptions that PrEP is only appropriate for certain groups, and emphasized the importance of community engagement. Several respondents said that community sensitization and stakeholder engagement played a critical role in establishing sites as safe spaces in

the community and promoting client engagement with the healthcare system. One 24-year-old male program manager from a Migori-based CSO whose program primarily served AGYW, described the important role of community leadership in bringing parents on board, saying that "[community leaders] come in and tell us 'you are conducting these kinds of meetings but we have not been informed. If only you could inform us, we could be sensitizing the [AGYW's] parents'." Others discussed opportunities for PrEP programs to collaborate with traditional circumcisers, social workers, members of key population communities, and religious leaders to develop communication strategies.

Program and county managers also saw community health workers or volunteers and community health assistants as critical for the success of PrEP programs, and emphasized the need for coordination and collaboration with these groups:

"Those are the people who are good at health promotion. Service providers . . . may find it a bit difficult. You see from the community health volunteers, the community health assistants, and the public health officers, you will find it easier."– 24-year-old male Program Manager, Migori-based CSO

**Advocacy.** Finally, all respondents acknowledged the important role of stigma in PrEP implementation, and saw a role for themselves and the wider community in PrEP advocacy. Program and county managers discussed the influence of stigma in the community and in the health care system:

"What we need to do is to intensify sensitization at community level and also with the health care workers. You know, there are health care workers who don't embrace [PrEP]. They always ask why are we doing that, and [saying] 'we know it is wrong, why are you exposing yourself so that you take PrEP?'"– 44-year-old male county manager

Many peer educators emphasized the importance of widespread community sensitization around PrEP, saying that they themselves faced stigma for the work they do. They discussed the community's perception of their work, saying "they say I promote immorality" and they face "discouragement" from others in the community. However, respondents also noted the role of stigma within the healthcare system. Many respondents discussed stigmatizing beliefs among PrEP service providers and other clinic staff, such as the belief that by providing PrEP "you are promoting promiscuity" or that "these guys [PrEP clients] are immoral and it is not supposed to be like this." In response to this challenge, many program and county managers emphasized the need for community awareness creation and improved provider training and sensitization.

## Monitoring and evaluation

Most respondents were enthusiastic about the value of routine data collection and review. Several expressed a desire for more regular opportunities to review data, along with more continuous supervision and support to do so. Several PrEP service providers said that, although it did not always fall within their job duties, they wanted to be more involved in data review meetings, in order to improve their approach and provide insights from client interactions. One suggested that participating in such meetings would motivate providers:

"Let partners accelerate the issue of monthly data review meetings, so that we can have intensified meetings. This will also help us to see where we are, where we have been, and how do we improve on some areas, especially on retention. Let it be sustainable, not having

meetings once and then it goes. That will be part of the motivation [for providers]."– 44-year-old male clinical officer, Migori

**Data collection tools.** PrEP service providers had mixed experiences with data collection tools, particularly the risk assessment screening tool. Some were familiar with the tool and had no concerns, saying things like "it makes my work easier" and describing it as "friendly" to use. Others were aware of the tool, but had concerns about its usability from both a provider and client perspective.

A few PrEP service providers expressed that the questions on the risk assessment screening tool might be uncomfortable for clients to answer; for example, a 41-year-old female nursing officer from Mombasa said that "in exit interviews, clients say the risk assessment screening form is so detailed and personal, so they were complaining." Another, a 27-year-old male HIV testing services provider from Migori, discussed the need to "equip our staff with all risk assessment screening questions and help us in constructing questions with the client" to improve both provider and client comfort with the questions in the tool.

*Monitoring & evaluation needs.* Despite general enthusiasm for program monitoring, a few respondents discussed challenges in this area. Some respondents said that data review meetings could be discouraging if not paired with solutions and support. For example, a 33-year-old female PrEP service provider from a DICE said, "the review meetings are good but sometimes. . . you will feel they have spoiled your whole day." The same provider went on to say that "they expect those review meetings to be supportive supervision," expressing frustration that those organizing the meetings did not appreciate the day-to-day realities of PrEP service delivery and their impact on the data. Other PrEP service providers felt that the amount of documentation involved in their day-to-day tasks was burdensome. In one FGD, providers discussed that HIV-related activities had a reputation of being difficult due to the paperwork burden:

"Most of the health workers don't want anything dealing with HIV. They think it is a burden. It increases their work load because there is also a lot of documentation, counseling and follow-up of clients, viral load, CD4, tuberculosis screening. It involves a lot of documentation so that is why they are afraid of the [HIV programs]."– 26-year-old male nursing officer, Mombasa

Respondents across groups also discussed facing pressure to meet PrEP program targets. Providers and peer educators in particular were concerned that targets might hinder their ability to provide quality services. Some peer educators discussed feeling discouraged when they failed to meet targets. A 45-year-old female peer educator from Mombasa said, "This target thing has destroyed everything." Another, a 22-year-old male peer educator from Migori expressed, "We try our best to mobilize, but people take a long time to accept the initiative. The clinic does not appreciate our efforts."

Program and county managers similarly described how pressure to meet targets influenced services. According to one 34-year-old program manager from a Nairobi-based CSO, while programs are striving to meet targets, they may miss gaps in services, such as recruitment of ineligible PrEP clients or eligible clients remaining uninitiated due to overstretched PrEP service providers. The manager suggested that more regular data review meetings with the entire PrEP team could be a solution: "Those quarterly reviews. . . that concept would have helped a lot to curb those small things that were happening." For managers, program monitoring

presented an opportunity to revisit progress and reiterate program goals, which they felt should not be limited to program leadership.

## Financing and resource mobilization

**Program and facility financing.** Respondents across groups described key financial considerations faced by PrEP programs and health facilities. Program and county managers highlighted the unique financial hurdles related to new programs. For example, one respondent discussed how there were inefficiencies early in the program, because they were not accustomed to a sub-contract mechanism:

> "It was something very new to us. We were not well oriented and you are told do 'milestones.' What is milestones?. . . We are not used to such kind of stuff. I think for me it's basically there was no orientation. . . Then there was those back and forth, coz you would present something, you are told you haven't done this, repeat, put this, come back again. . . It wasn't efficient."– 34-year-old male director, Nairobi-based CSO

Other financial constraints faced at the facility level included lack of vehicles, inadequate transport reimbursement or per diem during trainings, and lack of airtime for client follow-up. For example, many PrEP service providers described using their own mobile phone airtime to contact PrEP clients, but felt that paying for this was the responsibility of the facility.

Most respondents said that payment of PrEP program staff, particularly PrEP service providers and peer educators, was a critical issue influencing service delivery. Where increasing payment to staff was not feasible, some respondents suggested offering additional incentives, such as branded merchandise, may help staff feel more connected to the program and alleviate tensions related to finances. Peer educators themselves said they were passionate about their work and often willing to do it for minimal pay. However, they also expressed that increasing their payment would boost their morale and motivate them to do more for the program. In particular, peer educators felt that they would benefit from transportation vouchers or mobile phone airtime in addition to their monthly stipend, given the time spent accompanying clients to clinics or calling them to follow up:

> "Peer educators are paid 3,500 Ksh per month. You are not given transport reimbursements. You are told you have ten peers [clients] and they should come every three months. You should at least be given a transport voucher worth 2,000 Ksh per month. Then you will have that morale."– 39-year-old female PE, Mombasa

Respondents saw payment of program staff not only as an issue of morale, but linked it to client experiences as well. For example, one PrEP service provider from Mombasa explained how failure to sufficiently pay peer educators could change PrEP program impact:

> "That's why I have been saying that the peer educators are very important, because once they relay the information that 'I won't take him [client] back because I haven't been paid,' that's it. I was also working with [organization] and that is the same story that happened there. . . Once a peer educator is not paid, they fail to mobilize and won't come for screening. They would rather go to other facilities."– 38-year-old male clinical officer, Mombasa

**User financing.** Some respondents also discussed the role of financing at the client level. Although PrEP was provided for free, respondents were aware of opportunity costs related to

PrEP, such as time spent accessing services or dealing with side effects. In particular, respondents noted specific opportunity costs for FSW clients:

> "Some [FSW] are saying if they use the drug maybe in the evening and she was preparing to leave they sleep and forget. . . she sleeps and when she wakes up it's already morning and she was to go to work so they complain saying the drug gives sleep [and makes her lose business]."– 26-year-old-female PE, Mombasa

Client bus fare to and from clinics was an important consideration raised by respondents across groups. Respondents explained that, to initiate PrEP or refill a prescription, clients sometimes had to visit the facility multiple times, which increased costs. A 24-year-old female peer educator from Migori said that "this discourages them, because maybe she has gone back home and she doesn't want to come back because fare is a problem." Another respondent, a 33-year-old female PrEP service provider from a Nairobi-based DICE, said that fare becomes an issue when following up with PrEP clients: "Sometimes you have to follow them up. They will tell you, 'Today I don't have fare. I will come tomorrow.'"

While most respondents agreed that user financing created challenges for uptake and continued use of PrEP, a few respondents discussed how fee-for-service models could sustain PrEP programs in the long term. For some respondents, such a model could have broader implications for how clients approach their own healthcare:

> "If you want tea or *mandazi*, you pay, isn't it? You fill your stomach. Why won't you pay for Panadol (paracetamol) to deal with your headache? Something small? It should not always be free, free, free. . . We pay for things that we value. These free things we don't [value]. Let's shift our mind, small [fee], not so much, but something that will make me feel I have contributed to my health."– 50-year-old female Program Manager, Nairobi-based CSO

## Discussion

This framework analysis used the PIF to summarize PrEP implementation challenges and opportunities from a health system perspective. We moved beyond the end user perspective to identify broader considerations that have implications for PrEP scale-up in Kenya and more widely in SSA. Specifically, our analysis underscores the importance of PrEP program leadership and governance, carefully planned service delivery, training and capacity-building, and communication. While less-commonly discussed by respondents, we also identified important considerations around commodities, financing, and monitoring and evaluation.

We found that respondents prioritized leadership and governance, and saw successful leadership as clear communication, coordination, skills, inspiration, and championship at multiple levels and fostered opportunities for staff to share their opinions and experiences. This aligns with previous research, which has shown that strong leadership and coordination not only improve program efficiencies but reduce provider burnout [10, 28, 29]. From the findings, leadership challenges arose when staff felt they themselves or the PrEP program were insufficiently prioritized within the health system. Indeed, many peer educators and providers were passionate about PrEP and wanted to take on leadership in their own programs. This included a desire to be more involved in program monitoring and evaluation processes. Frontline PrEP service providers, given their role, are often more familiar with the feasibility and acceptability of implementation processes, and as such should be included more explicitly in leadership and decision-making.

PrEP is still a relatively new HIV intervention in SSA, and our findings reflect the broader challenges of rapidly scaling up a new intervention within a constrained health system,

including challenges related to human resources, financing, commodities, and organization of service delivery [12]. While financing issues varied and were often not discussed by frontline staff, we identified several issues related to staff payment and provision of resources (e.g. mobile phone airtime). Others have similarly found that issues related to pay can negatively impact HIV program staff morale [29]. In terms of commodity security and supply chain management, while supply of PrEP medication itself was a rare challenge, we identified some concerns related to commodities complementary to PrEP, such as infrequent stock-outs HIV test kits, condoms and lubricants. Because clients accessing PrEP should receive a comprehensive HIV prevention package, programs should consider how increased demand for PrEP might pose challenges for stocking these complementary items. Commodity supply may also have implications for morale, as staff may feel frustrated or guilty for lack of facility resources [29].

Despite PrEP's nascency in SSA, we found that staff felt well-trained to deliver services. However, they had concerns about increased workloads and burnout, particularly in contexts where PrEP was integrated with other services. This mirrors staffing challenges identified in other PrEP and ART settings, where increased workload and burnout resulting from staff shortages have been found to not only perpetuate human resource challenges, but also reduced quality of services [18, 20, 30]. Our findings show that burnout was a particular concern for PrEP service providers who worked in integrated programs, where they were called to prioritize multiple competing responsibilities, such as provision of family planning and PrEP. Because of this, some respondents preferred models where PrEP was offered as a standalone service, rather than integrated into other HIV or sexual and reproductive health services. Studies have highlighted the important opportunities provided by integration of PrEP with other services, including improved PrEP access and linkage to other important health services to further reduce health inequities [16, 31]. However, our findings indicate that integration should be implemented carefully and with special consideration for its acceptability and impact on client experiences, including wait times and stigma.

We identified a need for coordinated communication strategies, including within the health system and between the health system and communities. This included more tailored demand creation and dedicated community engagement efforts. Others have discussed the importance of these efforts in the context of PrEP and other HIV programs [16, 32, 33]. Within the health system, we found that respondents valued effective communication at all levels and stages of PrEP service delivery. Communication about program data and progress was particularly important to respondents in our study. We found that PrEP service providers and peer educators were interested in contributing to monitoring and evaluation, understanding how targets were set, and using data to inform broader program decisions. Where possible, programs may want to involve broader implementation teams in monitoring and evaluation processes, beyond routine data collection. This could not only ensure on-the-ground perspective is brought into broader program decisions, but could provide staff with opportunities for capacity building and leadership, in turn fostering high-quality service delivery and program longevity.

There are some limitations to our study. First, due to the broad nature of our guides, we were unable to discuss all issues in-depth with each respondent or group. However, this allowed us to cover a breadth of topics with respondents, and we were able to achieve saturation of the themes presented. Second, our approach to framework analysis was highly deductive in nature, which may have precluded us from identifying additional themes not included in the PIF. However, because the PIF is fairly comprehensive in nature, we felt comfortable with a deductive approach. We also included an "other" code and used memos to identify unanticipated themes. Finally, our recruitment method may have influenced the types of respondents included in our study. Because recruitment was purposive and via facility in-

charges, it may be that some individuals felt uncomfortable sharing negative feedback about their facilities or that certain individuals were unwilling to participate. Despite this possibility, we were able to identify a variety of concerns and challenges, as well as positive experiences with PrEP programs.

Beyond implementation considerations, our findings highlight key areas for additional research. First, it is important to more thoroughly explore the impact of service integration from both a health system and user perspective. Our findings highlight important concerns about integrating PrEP with other services. Better understanding how these concerns directly impact client experiences and program quality and efficiency will be important as programs continue to scale up in SSA. Second, greater understanding of task-shifting/sharing within PrEP service delivery is warranted. Our findings about PrEP service provider burnout and task prioritization reinforce the findings of others [18, 20]. Future research should examine the impacts of task-shifting/sharing on implementation and client outcomes [34]. Finally, our findings extend only to implementation of daily oral PrEP. Additional research will be needed on health system considerations for delivering new dosing regimens.

## Conclusion

In the context of a nationally-scaled PrEP program, we found that the domains highlighted in Kenya's PrEP Implementation Framework were thought to enable delivery of PrEP services with efficiency and coordination. This was reflected in PrEP service providers', peer educators', and program and county managers' value of leadership, communication and coordination, and expedient use of data. However, concerns about service integration and service delivery in resource-constrained settings need to be addressed. Further, findings suggest that Kenya's PrEP Implementation Framework aligns closely with the priorities of individuals involved in PrEP service delivery and management. Other programs should consider the potential role of similar implementation frameworks to guide PrEP scale-up.

## Supporting information

**S1 File.**
(DOCX)

## Acknowledgments

We would like to acknowledge the contributions of Jilinde staff who participated in instrument development and data collection, transcription, and translation of data. We also thank respondents for their time and contributions to this study.

## Author Contributions

**Conceptualization:** Kaitlyn Atkins, Abednego Musau, Mary Mugambi, Soud Ali Tengah, Mercy Kamau, Ruth Kumau, Jason Reed, Daniel Were.

**Formal analysis:** Kaitlyn Atkins.

**Funding acquisition:** Daniel Were.

**Investigation:** Abednego Musau, Geoffrey Odhyambo, Daniel Were.

**Methodology:** Abednego Musau, Daniel Were.

**Project administration:** Abednego Musau, Geoffrey Odhyambo, Soud Ali Tengah, Mercy Kamau, Ruth Kumau, Daniel Were.

**Supervision:** Abednego Musau, Mary Mugambi, Geoffrey Odhyambo, Jason Reed, Daniel Were.

**Writing – original draft:** Kaitlyn Atkins.

**Writing – review & editing:** Kaitlyn Atkins, Abednego Musau, Mary Mugambi, Geoffrey Odhyambo, Soud Ali Tengah, Mercy Kamau, Ruth Kumau, Jason Reed, Daniel Were.

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
