## [Decision Letter · Decision Letter 0]

13 Jul 2021

PONE-D-21-13431

Health system opportunities and challenges for PrEP implementation in Kenya: A qualitative framework analysis

PLOS ONE

Dear Dr. Atkins,

Thank you for submitting your manuscript to PLOS ONE. After careful consideration, we feel that it has merit but does not fully meet PLOS ONE’s publication criteria as it currently stands. Therefore, we invite you to submit a revised version of the manuscript that addresses the points raised during the review process.

We look forward to receiving your revised manuscript.

Kind regards,

Rupa R. Patel, MD

Academic Editor

PLOS ONE

Journal Requirements:

2. Please provide additional details regarding participant consent. In the ethics statement in the Methods and online submission information, please ensure that you have specified whether consent was informed.

3. Please include a copy of the interview guide used in the study, in both the original language and English, as Supporting Information, or include a citation if it has been published previously.

Additional Editor Comments:

The authors have assessed a much needed topic to aid PrEP scale up.

In addition to the reviewers’ recommendations, other suggested manuscript revisions are the following:

1. There are several acronyms in the paper and it is unclear how they were chosen.

2. Please see the reviewer comments regarding strengthening the content in this manuscript regarding commodity security.

3. Please provide more context for the reader regarding the different facilities (i.e., DICE, CCC, and HIV treatment centers) in the methods section. For example, the authors have brought in new acronyms late into the paper and later into the results section (i.e., Service Delivery section and CCC).

4. For the results section, please organize the text into Themes and Sub-Themes.

5. Please provide a figure of the PIF being used for this study.

6. Please provide the qualitative guides for FDGs and informant interviews of all groups as appendices.

7. Please provide more standardization for presenting of quotes in this paper. Suggestions include formatting the quotes so prominent ones are more visible, as well as uniformly labelling each quote throughout the paper with the use of age, gender, setting type, and location.

8. Under the service delivery section, what is “outreaches.” Please provide more context in the methods section.

9. In the first line of the conclusion, the authors are over inflating the study findings to suggest that PIF enabled delivery of PrEP services “with efficiency and coordination.” The study and its sampling was not designed to nor can robustly assess these factors. In addition, the study findings did not strongly suggest this.

Reviewers' comments:

Reviewer's Responses to Questions

**Comments to the Author**

1. Is the manuscript technically sound, and do the data support the conclusions?

Reviewer #1: Yes

Reviewer #2: Yes

2. Has the statistical analysis been performed appropriately and rigorously? 

Reviewer #1: Yes

Reviewer #2: Yes

3. Have the authors made all data underlying the findings in their manuscript fully available?

Reviewer #1: Yes

Reviewer #2: Yes

4. Is the manuscript presented in an intelligible fashion and written in standard English?

Reviewer #1: Yes

Reviewer #2: Yes

5. Review Comments to the Author

Reviewer #1: This manuscript described a qualitative study exploring the experiences of PrEP service providers, peer educators, and program or county managers in Kenya in order to better understand how their PrEP delivery approach aligns with Kenya’s 2017 National PrEP implementation Framework. The study appears to be relevant to the aims of PLOS ONE in that it provides new insights into PrEP implementation and service delivery practices and potential gaps that can be addressed or strengthened. The study contains minor areas for improvement.

MINOR COMMENTS

• The authors might consider removing Table 3 or combining it with Table 2, much of the information provided is already in-text and/or in Table 2.

• In the results section, the authors may consider shortening the exemplar quotations.

• The authors highlighted sub-codes in Table 1. However, in the results section it is not clear which quotations support which sub-code. Considering that sub-codes are highlighted in Table 1, the authors may add sub-headers in the results section to further highlight these sub-codes and their corresponding quotations.

• The discussion section feels a bit incomplete. While the authors present the implications of results related to Leadership and Governance, Service Delivery, and Communications, Advocacy and Community Engagement, there appears to be little or no discussion about other Parent Codes such as Commodity Security and Supply Chain Management, Monitoring and Evaluation, and Financing and Resource mobilization. This may reflect an opportunity to select the most relevant and important Parent Codes and focus the manuscript on those codes. This will help tighten and focus the manuscript significantly.

• One very minor comment is that there were many non-standard acronyms.

Reviewer #2: This is a great and timely article and focuses on one of the key areas in PrEP implementatoin often neglected, e.g. what happens after full scale up of PrEP is complete. The authors condutcted a thorough study of Kenya's PrEP implementation system grounded in the Kenyan plan for implementation. The article was conducted in a rigourous manner.

Abstract: In the RESULTS section, there is a sentence about "commodity security" not be strongly priorotitized. Reading the full results section of the manuscript, it actually seems like participants think this needs to be prioritized more. I think this is more of a wording issue and that this might need to be rephrased as it is in the same sentence where research is not strongly prioritized (which I agree with).

Background: This is well written. I would have liked to see a brief explanation of the difference between DICE, CCC, and HIV treatment centers here (or elsewhere) as when I started reading the last paragraph within the COMMODITY's section in the Results section, I realized I did not have a thorough understanding of these different facilities. I had assumed the CCC was a one-stop shop, but one of the statements in that paragraph alluded to it not being a one-stop show.

Methods: I would have liked a few more details as to why you kept the KII as the leaders in this setting and had everyone else in FGD. I wonder what a few KII with providers and/ peer educations would have provided.

Results: Throughout your results section, you need to separate out the subthemes. Your results can be slightly difficult to follow without subtheme headers. For example within sevice delivery the first paragraph can be location of services, second paragraph can be staff roles, etc.

Discussion: Overall this is well written. Recommend condensing the first 2 paragraphs as they both feel like summaries of the results. Also in the 2nd paragraph, the sentence starting with "Indeed, many peer educators..." was not addressed in the results section.

6. PLOS authors have the option to publish the peer review history of their article (what does this mean?). If published, this will include your full peer review and any attached files.

Reviewer #1: No

Reviewer #2: No

---

## [Author Response · Author response to Decision Letter 0]

20 Oct 2021

RESPONSE TO REVIEWERS

Reviewer #1

This manuscript described a qualitative study exploring the experiences of PrEP service providers, peer educators, and program or county managers in Kenya in order to better understand how their PrEP delivery approach aligns with Kenya’s 2017 National PrEP implementation Framework. The study appears to be relevant to the aims of PLOS ONE in that it provides new insights into PrEP implementation and service delivery practices and potential gaps that can be addressed or strengthened. The study contains minor areas for improvement.

Thank you for your feedback and for the opportunity to make these improvements.

1. The authors might consider removing Table 3 or combining it with Table 2, much of the information provided is already in-text and/or in Table 2.

Thank you for this suggestion. We have consolidated these into a single table (Table 2).

2. In the results section, the authors may consider shortening the exemplar quotations.

We have reviewed all exemplar quotations for length and redundancy. Where possible without loss of meaning, we have shortened quotations.

3. The authors highlighted sub-codes in Table 1. However, in the results section it is not clear which quotations support which sub-code. Considering that sub-codes are highlighted in Table 1, the authors may add sub-headers in the results section to further highlight these sub-codes and their corresponding quotations.

Thanks for this comment. We regret the use of the sub-codes was not made clear in our first draft. Sub-codes do not align directly with specific quotations, as our coding approach was broad and many quotations aligned with two or more sub-codes. We have added language to the Methods section to clarify this. We have also added sub-headers in the results which align with many sub-codes to better orient readers.

4. The discussion section feels a bit incomplete. While the authors present the implications of results related to Leadership and Governance, Service Delivery, and Communications, Advocacy and Community Engagement, there appears to be little or no discussion about other Parent Codes such as Commodity Security and Supply Chain Management, Monitoring and Evaluation, and Financing and Resource mobilization. This may reflect an opportunity to select the most relevant and important Parent Codes and focus the manuscript on those codes. This will help tighten and focus the manuscript significantly.

Thank you for this suggestion. We did consider removing these domains from the manuscript altogether. However, given the goal of this manuscript was to assess the alignment and application of the PrEP Implementation Framework (PIF) with frontline workers’ and managers’ perspectives on PrEP implementation, we felt it was important to retain most PIF domains in our results (with the exception of research, which was largely unmentioned).

However, in response to this comment, we have added language to the discussion that we hope better captures implications related to these three important areas, while acknowledging they were less-commonly discussed by respondents.

5. One very minor comment is that there were many non-standard acronyms.

We have reviewed our usage of acronyms to ensure that acronyms are only used for phrases which are used repeatedly throughout the document (e.g. FSW, PrEP). For others (e.g. HTS, RAST, CHV, TWG), we have replaced acronyms with full-text phrases given their infrequency of use. We also hope the addition of Table 3 (in response to Reviewer 2, below) can clarify the acronyms referring to PrEP service delivery sites (e.g. CCC, DICE).

Reviewer #2

This is a great and timely article and focuses on one of the key areas in PrEP implementation often neglected, e.g. what happens after full scale up of PrEP is complete. The authors conducted a thorough study of Kenya's PrEP implementation system grounded in the Kenyan plan for implementation. The article was conducted in a rigorous manner.

Again, we are grateful for your feedback and for the opportunity to improve the manuscript.

1. Abstract: In the RESULTS section, there is a sentence about "commodity security" not be strongly prioritized. Reading the full results section of the manuscript, it actually seems like participants think this needs to be prioritized more. I think this is more of a wording issue and that this might need to be rephrased as it is in the same sentence where research is not strongly prioritized (which I agree with).

Thanks for this important clarification. What we meant by this sentence (as previously written) was that respondents did not emphasize these topics during interviews. However, the reviewer is correct that respondents certainly prioritized commodity issues. We have rephrased this in an effort to clarify.

2. Background: This is well written. I would have liked to see a brief explanation of the difference between DICE, CCC, and HIV treatment centers here (or elsewhere) as when I started reading the last paragraph within the COMMODITY's section in the Results section, I realized I did not have a thorough understanding of these different facilities. I had assumed the CCC was a one-stop shop, but one of the statements in that paragraph alluded to it not being a one-stop show.

This is an important point, and one we regret to have previously overlooked. We have inserted a table (now Table 3) in the Results section, which delineates the service delivery platforms described by respondents.

3. Methods: I would have liked a few more details as to why you kept the KII as the leaders in this setting and had everyone else in FGD. I wonder what a few KII with providers and/ peer educations would have provided.

We agree that conducting KIIs with providers or peer educators might have provided us with in-depth, individual experiences and perspectives. The goal of our data collection with providers and PEs was, broadly, to understand attitudes and experiences at a group level, and to explore areas of homogeneity and heterogeneity within these groups. As such, we chose FGDs given their suitability for qualitative research with these aims. Further, our experience working with these groups shows that FGDs can provide important information about group interactions, which yield unique insights not obtainable through one-on-one interviews. 

KIIs were used for program managers in part to accommodate challenges with logistics; coordinating focus groups with managers may have yielded important insights, but was infeasible given their wide geographic distribution and other commitments. 

4. Results: Throughout your results section, you need to separate out the subthemes. Your results can be slightly difficult to follow without subtheme headers. For example within service delivery the first paragraph can be location of services, second paragraph can be staff roles, etc.

Thanks for this comment. In response to this and a similar comment from Reviewer 1, we have inserted subheadings to align (where appropriate) with sub-codes. We hope this improves the flow of the Results.

5. Discussion: Overall this is well written. Recommend condensing the first 2 paragraphs as they both feel like summaries of the results. 

Thank you. Overall, we have worked to streamline the discussion to eliminate redundancy and condense summaries where possible.

6. Also in the 2nd paragraph, the sentence starting with "Indeed, many peer educators..." was not addressed in the results section.

Thank you for flagging this. The first part of this sentence was in reference to the role providers and peer educators saw for themselves in leadership (discussed at the end of the first theme in Results). However, the reviewer correctly noted that the latter half of this sentence (“but did not always feel supported to do so”) was not addressed in the results. As such, we have removed this portion of the sentence.

RESPONSE TO EDITOR SUGGESTIONS

1. There are several acronyms in the paper and it is unclear how they were chosen.

We believe this has been addressed through our response to Reviewer 1 and removal of several acronyms.

2. Please see the reviewer comments regarding strengthening the content in this manuscript regarding commodity security.

Thank you; we believe this has now been addressed in response to comments from both reviewers. 

3. Please provide more context for the reader regarding the different facilities (i.e., DICE, CCC, and HIV treatment centers) in the methods section. For example, the authors have brought in new acronyms late into the paper and later into the results section (i.e., Service Delivery section and CCC).

This has been added; see Table 3.

4. For the results section, please organize the text into Themes and Sub-Themes.

This has been done.

5. Please provide a figure of the PIF being used for this study.

Unfortunately, the PrEP Implementation Framework is not a traditional framework in the sense that it does not specify relationships between the key domains in the framework. Instead, it “aims to provide guidance on the roll out of PrEP in Kenya,” by specifying seven focus areas which, taken holistically, were thought to facilitate effective delivery and scale-up of PrEP. We regret that we are unable to insert a figure of the PIF because of this, but we rather provide a textual summary of the framework.

6. Please provide the qualitative guides for FDGs and informant interviews of all groups as appendices.

We are in the process of compiling the guides and creating the supplemental file. As several of our team members are away and we did not want to delay the resubmission process, we are submitting without these for the time being but will be sure to have the supplemental file/appendix ready to upload with the next iteration of the paper. Sincere apologies for the delay.

7. Please provide more standardization for presenting of quotes in this paper. Suggestions include formatting the quotes so prominent ones are more visible, as well as uniformly labelling each quote throughout the paper with the use of age, gender, setting type, and location.

We have reviewed formatting of quotes for consistency. Per the editor’s suggestion, we made more prominent quotes visible by offsetting (indenting) them. We have also reviewed labelling and note that all quotes are labeled with age, gender, participant type, and location (either when introducing the quote if in-text, or as an attribution for offset quotes). We note that an exception was made for county managers, whom are easily identifiable by their age, gender, and location. For example, we stated “A 44-year-old male county manager” rather than describing the location of the county manager, because in many instances there is only one county manager in a given location. On the other hand, program managers and providers are less easily identifiable by their age or location, so we provided all details for these individuals.

8. Under the service delivery section, what is “outreaches.” Please provide more context in the methods section.

This has been added; see Table 3.

9. In the first line of the conclusion, the authors are over inflating the study findings to suggest that PIF enabled delivery of PrEP services “with efficiency and coordination.” The study and its sampling was not designed to nor can robustly assess these factors. In addition, the study findings did not strongly suggest this.

Thank you for this feedback. We agree that, as a qualitative study, this study was not designed to quantify efficiency or coordination of service delivery. After review of the statement as previously written, we realize that it erroneously implied the PIF itself enabled this efficiency and coordination. We have revised the sentence as follows (changes in all caps): “…we found that THE DOMAINS HIGHLIGHTED IN Kenya’s PrEP Implementation Framework WERE THOUGHT TO enable delivery of PrEP services with efficiency and coordination.” We hope that this underscores the qualitative nature of our research and avoids overstating findings, but defer to the editor if further adjustments are warranted.

---

## [Editor Report · Decision Letter 1]

26 Oct 2021

Health system opportunities and challenges for PrEP implementation in Kenya: A qualitative framework analysis

PONE-D-21-13431R1

Dear Dr. Atkins,

We’re pleased to inform you that your manuscript has been judged scientifically suitable for publication and will be formally accepted for publication once it meets all outstanding technical requirements.

Kind regards,

Rupa R. Patel, MD

Academic Editor

PLOS ONE

---

## [Editor Report · Acceptance letter]

28 Sep 2022

PONE-D-21-13431R1 

Health system opportunities and challenges for PrEP implementation in Kenya: A qualitative framework analysis 

Dear Dr. Atkins:

I'm pleased to inform you that your manuscript has been deemed suitable for publication in PLOS ONE. Congratulations! Your manuscript is now with our production department. 

Kind regards, 

on behalf of

Dr. Rupa R. Patel 

Academic Editor

PLOS ONE